# Pushing Toward the Simplex Vertices: A Simple Remedy for Code Collapse in Smoothed Vector Quantization

## Abstract

Vector quantization, which discretizes a continuous vector space into a finite set of representative vectors (a *codebook*), has been widely adopted in modern machine learning. Despite its effectiveness, vector quantization poses a fundamental challenge: the non-differentiable quantization step blocks gradient backpropagation. *Smoothed* vector quantization addresses this issue by relaxing the hard assignment of a codebook vector into a weighted combination of codebook entries, represented as the matrix product of a simplex vector and the codebook. Effective smoothing requires two properties: (1) smoothed quantizers should remain close to a onehot vector, ensuring tight approximation, and (2) all codebook entries should be utilized, preventing *code collapse*. Existing methods typically address these desiderata separately. By contrast, the present study introduces a simple and intuitive regularization that promotes both simultaneously by minimizing the distance between each simplex vertex and its $K$-nearest smoothed quantizers. Experiments on representative benchmarks—including discrete image autoencoding and contrastive speech representation learning—demonstrate that the proposed method achieves more reliable codebook utilization and improves performance compared to prior approaches.

## 1 Introduction

Vector quantization is a method for discretizing a continuous vector space (Gray, 1984; van den Oord et al., 2017). It maps each vector in the continuous space to the nearest element of a finite set of representative vectors (a.k.a. a *codebook*). The resulting discrete representations are easier to manipulate and interpret than the original continuous forms, and have proven effective across diverse applications, including image generation (Esser et al., 2021; Ramesh et al., 2021; Rombach et al., 2022; Yu et al., 2022b), speech recognition (Baevski et al., 2020a;b), and music generation (Hadjeres & Crestel, 2020; Dhariwal et al., 2020).

When integrated into deep neural networks, however, vector quantization introduces a fundamental challenge: quantization is a non-differentiable operation that blocks gradient backpropagation (van den Oord et al., 2017). Accordingly, some approximation is required to enable learning through quantization.

One effective workaround is to *smooth* vector quantization (Jang et al., 2017). The selection of a codebook vector can be expressed as multiplying the codebook matrix by its corresponding onehot vector $(0, \ldots, 1, \ldots, 0)^\mathsf{T}$, whose nonzero entry indexes the chosen vector. *Smoothed* vector quantization relaxes this onehot vector to lie within the simplex $\Delta^{M-1} := \{(p_1, \ldots, p_M) \mid \sum_{m=1}^{M} p_m = 1\}$, where $M$ denotes the number of codebook vectors. Consequently, differentiable mappings (e.g., softmax) become available and can be incorporated into neural networks.

To successfully approximate onehot quantizers, smoothed quantizers must be distributed around the vertices of the simplex (orange "o" in Figure 1A). At the same time, they should not concentrate around a few vertices, leaving other codebook entries unused (blue "x"). This latter issue—called *code collapse*—has been identified as a major challenge in vector quantization (Dieleman et al.,

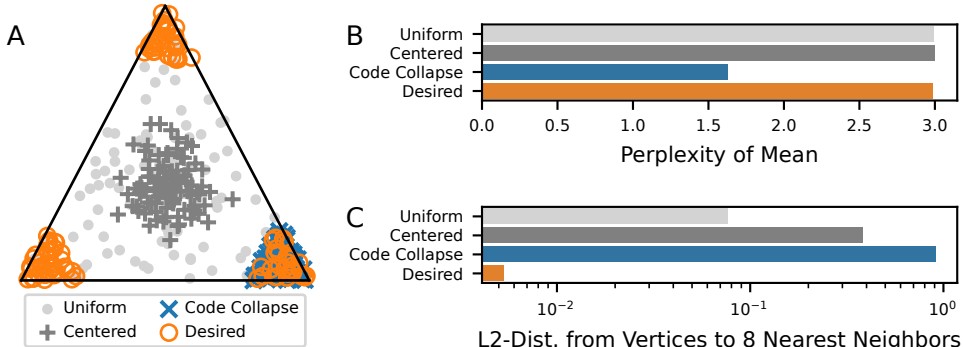

Figure 1: (A) Four different distributions on the simplex $\Delta^{3-1}$. For effective smoothed vector quantization, samples should be concentrated near the vertices of the simplex (i.e., onehot-like vectors; orange), rather than centered (dark gray) or uniformly spread across the simplex (light gray). At the same time, each vertex must be neighbored by some samples to avoid code collapse (blue). (B) Maximizing the perplexity of the sample mean (Baevski et al., 2020b) penalizes code collapse but cannot discriminate among the other three distributions. (C) The proposed $K$-nearest neighbor (KNN) distance minimization ($K = 8$) favors the desired vertex-concentrated distribution while also preventing code collapse.

2018; Baevski et al., 2020b; Dhariwal et al., 2020; Fifty et al., 2025). A widely adopted workaround for code collapse (in smoothed quantization) is to introduce an auxiliary learning objective that maximizes the entropy or perplexity of the *mean* of the smoothed quantizers (Dieleman et al., 2018; Baevski et al., 2020b, see §2.3 for the formal definition). However, maximizing the entropy/perplexity of the mean can be achieved by various distributions, not only the desired vertex-neighboring ones. For example, both uniform and center-concentrated distributions have their mean at the simplex center (Figure 1A), which also maximizes this objective (Figure 1B). Accordingly, an additional mechanism is needed to tighten the smoothing (e.g., by adjusting the temperature parameter of the (Gumbel-)softmax; see §2.2).

Beyond this standard approach, however, there exists a simple and unified strategy for simultaneously tightening smoothed quantization and maximizing codebook usage: *Why don't we directly encourage clustering around all simplex vertices?* Specifically, minimizing the distance between each simplex vertex and its $K$-nearest neighbors (KNNs) satisfies both desiderata at once (Figure 1C). The present study investigates this intuitive yet underexplored approach, comparing it against existing alternatives on representative benchmarks. The results indicate that the proposed method enables the exhaustive usage of the entire codebook, even when other approaches suffer from code collapse.

The contributions of this work are summarized as follows:

- Neural vector quantization is reformulated as a smoothing problem of onehot vectors. This simple reformulation has been absent in the literature, which traditionally framed vector quantization as an extension of variational autoencoding (Kingma & Welling, 2014; Jang et al., 2017; van den Oord et al., 2017).

- Under this reformulation, an effective regularization loss function is proposed to promote both tight smoothing and exhaustive code utilization. The method demonstrates robustness across different learning settings.

The remainder of this paper is organized as follows. §2 reviews related studies on vector quantization. §3 introduces the proposed method, which is then evaluated on representative benchmarks in §4. Finally, §5 discusses the results and the limitations of the proposed method.

## 2 RELATED STUDIES

The central challenge of vector quantization lies in its non-differentiability, which disrupts the back-propagation of gradients in neural networks. To address this, either the gradient computation (backward path) or the quantization itself (forward path) must be approximated.

### 2.1 APPROXIMATION IN THE BACKWARD PATH (GRADIENT ESTIMATION)

One line of work retains the original non-differentiable quantization in the forward path but replaces the gradient computation in the backward path. Let $\mathbf{z} \in \mathbf{R}^D$ denote a pre-quantized feature vector, and $\{\mathbf{q}_1, \dots, \mathbf{q}_M\} \subset \mathbf{R}^D$ the set of quantized vectors. Quantization maps $\mathbf{z}$ to its "closest" codebook entry according under a distance metric $\mathcal{D}$: i.e., $\mathbf{z} \mapsto \mathbf{q}_{\iota(\mathbf{z})}$ where $\iota(\mathbf{z}) := \operatorname{argmin}_m \mathcal{D}(\mathbf{z}, \mathbf{q}_m)$. Consequently, the partial derivatives $\frac{\partial q_{\iota(\cdot),i}}{\partial z_j}$ are ill-defined and must be approximated.

A canonical approximation—known as the *straight-through estimation* (STE)—replaces the ill-defined Jacobian with the identity matrix (van den Oord et al., 2017):

$$\frac{\partial q_{\iota(\cdot),i}}{\partial z_j} \approx \begin{cases} 1 & i = j \\ 0 & \text{otherwise} \end{cases} \tag{1}$$

Algorithmically, the STE is implemented using the `detach` operation, which excludes its argument from gradient computation:

$$\text{STE}(\mathbf{q}_{\iota(\mathbf{z})}, \mathbf{z}) := \mathbf{q}_{\iota(\mathbf{z})} + \mathbf{z} - \texttt{detach}(\mathbf{z}) \tag{2}$$

Here, $z - \texttt{detach}(z)$ evaluates to zero, while the gradient with respect to $z$ can still be propagated through the first term.

More recently, Fifty et al. (2025) proposed an alternative gradient approximation, and demonstrated its empirical superiority over the STE:

$$\text{RE}(\mathbf{q}_{\iota(\mathbf{z})}, \mathbf{z}) := \texttt{detach}\left(\frac{\|\mathbf{q}_{\iota(\mathbf{z})}\|}{\|\mathbf{z}\|}R\right)\mathbf{z} \tag{3}$$

where $R$ is the rotation matrix aligning $\mathbf{z}$ to $\mathbf{q}_{\iota(\mathbf{z})}$,[1] and $\frac{\|\mathbf{q}_{\iota(\mathbf{z})}\|}{\|\mathbf{z}\|}$ rescales the rotated vector to match the amplitude of $\mathbf{q}_{\iota(\mathbf{z})}$. In this formulation, the Jacobian of the quantization is approximated by the scaled rotation matrix:

$$\frac{\partial q_{\iota(\cdot),i}}{\partial z_j} \approx \frac{\|\mathbf{q}_{\iota(\mathbf{z})}\|}{\|\mathbf{z}\|}R \tag{4}$$

### 2.2 APPROXIMATION IN THE FORWARD PATH (SMOOTHING)

An alternative approach approximates the forward quantization itself. Using the onehot representation $\mathbf{e}_m$ of the code index $m$, quantization can be expressed as:

$$\mathbf{q}_{\iota(\mathbf{z})} = Q\mathbf{e}_{\iota(\mathbf{z})} \tag{5}$$

where $Q := (\mathbf{q}_1, \dots, \mathbf{q}_M)$. *Smoothed* quantization extends the possible range of $\mathbf{e}_m$ to the simplex $\Delta^{M-1}$. As noted in §1, effective learning requires smoothed quantizers $\mathbf{p} \in \Delta^{M-1}$ to concentrate near the vertices of the simplex (i.e., $\mathbf{p} \approx \mathbf{e}_m$ for some $m$).

A widely studied instance of smoothed quantization is Gumbel-softmax sampling (Jang et al., 2017). Given assignment probabilities $\pi_m$ of $\mathbf{z}$ to the $m$-th code—typically log-proportional to their dot-product $\mathbf{q}_m^\top \mathbf{z}$—categorical sampling can be implemented using Gumbel samples $g_m = -\log(-\log u_m)$ with $u_m \sim \text{Uniform}(0,1)$:

$$\iota(\mathbf{z}) \sim \text{Categorical}(\pi_1, \dots, \pi_M)$$
$$\Leftrightarrow \iota(\mathbf{z}) = \operatorname*{argmax}_m(g_m + \log \pi_m) \tag{6}$$

---

[1]The rotation matrix is given by $R = I - 2\hat{\mathbf{r}}\hat{\mathbf{r}}^\top + 2\hat{\mathbf{q}}_{\iota(\mathbf{z})}\hat{\mathbf{z}}^\top$, where $\hat{\mathbf{v}} := \mathbf{v}/\|\mathbf{v}\|$ is the L2-normalization of vector $\mathbf{v}$, and $\mathbf{r} := \hat{\mathbf{q}}_{\iota(\mathbf{z})} + \hat{\mathbf{z}}$.

Replacing $\mathrm{argmax}$ above with $\mathrm{softmax}$ yields a smoothed quantization:

$$p_m = \frac{\exp\left(\left(g_m + \log \pi_m\right)/\tau\right)}{\sum_{m'} \exp\left(\left(g_{m'} + \log \pi_{m'}\right)/\tau\right)} \tag{7}$$

where lowering the temperature parameter $\tau$ produces a tighter approximation of categorical sampling.

The Gumbel-softmax sampling can also be combined with hard quantization using the STE:

$$\mathrm{STE}(\mathbf{e}_{\iota(\mathbf{z})}, \mathbf{p}) = \mathbf{e}_{\iota(\mathbf{z})} + \mathbf{p} - \mathrm{detach}(\mathbf{p}) \tag{8}$$

## 2.3 REGULARIZATION

Beyond approximation strategies, vector quantization also requires auxiliary regularization losses to ensure effective training. For example, the STE alone does not guarantee alignment between pre-quantized features and codebook entries. This alignment is instead fostered by the following regularization loss, $\mathcal{L}_{\mathrm{reg}}$ (van den Oord et al., 2017; Fifty et al., 2025):

$$\mathcal{L}_{\mathrm{total}} = \mathcal{L}_{\mathrm{main}} + \mathcal{L}_{\mathrm{reg}} \tag{9}$$

$$\mathcal{L}_{\mathrm{reg}} = \mathcal{L}_{\mathrm{hard}} := N^{-1} \sum_{i=1}^{N} \left( \beta \underbrace{\left\| \mathbf{z}_i - \mathrm{detach}(\mathbf{q}_{\iota(\mathbf{z}_i)}) \right\|^2}_{\text{Commitment Loss}} + \underbrace{\left\| \mathrm{detach}(\mathbf{z}_i) - \mathbf{q}_{\iota(\mathbf{z}_i)} \right\|^2}_{\text{Codebook Loss}} \right) \tag{10}$$

where $\mathcal{L}_{\mathrm{main}}$ is the primary task loss (e.g., L2 regression in autoencoding), and $\beta > 0$ is a weighting hyperparameter. As defined in in Equation 10, $\mathcal{L}_{\mathrm{hard}}$ consists of two components. The first term (known as the *commitment loss*) aligns each pre-quantized feature $\mathbf{z}_i$ in a batch ($i = 1, \ldots, N$) with their nearest codebook entry $\mathbf{q}_{\iota(\mathbf{z}_i)}$. The second term (called the *codebook loss*) moves each codebook vector $\mathbf{q}_m$ toward the centroid of its assigned features whose nearest neighbor is $\mathbf{q}_m$ (i.e., $\{\mathbf{z}_i : \iota(\mathbf{z}_i) = m\}$).

It should be noted, however, that the codebook loss in Equation 10 does not inherently prevent code collapse. Some codebook vectors may never serve as the nearest neighbor of any pre-quantized feature and therefore receive no updates (Zhu et al., 2025). Accordingly, previous studies have resorted to additional workarounds; for example, unused codebook vectors may be reset to the positions of pre-quantized features Dhariwal et al. (2020). More recently, Zhu et al. (2025) proposed another remedy called *SimVQ*, which reparameterizes the codebook $Q$ as the product of a randomly frozen matrix $Q' \in \mathbb{R}^{M \times D}$ and a learnable matrix $W \in \mathbb{R}^{D \times D}$ (i.e., $Q = Q'W$). In this formulation, all codebook vectors share learnable parameters with one another ($\mathbf{q}_m = q'_{m,1}\mathbf{w}_1 + \cdots + q'_{m,D}\mathbf{w}_D$), so updates to one vector propagate to the others.

Similarly, smoothed quantization also requires auxiliary regularization to avoid code collapse. A widely adopted option is the normalized perplexity of the mean assignment probability (Dieleman et al., 2018; Baevski et al., 2020b):

$$\mathcal{L}_{\mathrm{reg}} = \mathcal{L}_{\mathrm{ppl}} := \frac{\exp\left(-\sum_m \bar{\pi}_m \log \bar{\pi}_m\right)}{M} \tag{11}$$

$$\bar{\pi}_m := N^{-1} \sum_{i=1}^{N} \pi_{i,m} \tag{12}$$

As noted in §1, this perplexity-based regularization does not promote the onehotness of $\boldsymbol{\pi}_i$, failing to distinguish onehot-like samples from uniform or centered ones (Figure 1B). Onehotness can instead be induced by annealing the temperature parameter of the Gumbel-softmax sampling ($\tau \to 0$). However, manually scheduling this annealing is empirically challenging. The next section therefore proposes an alternative regularization loss that automatically encourages onehotness within the framework of gradient-based learning, while simultaneously preventing code collapse.

## 3 METHODS

As noted in previous sections, ideal smoothed quantizers $\mathbf{p} \in \Delta^{M-1}$ are distributed near the vertices of the simplex (Figure 1A). Moreover, each vertex should have at least some smoothed quantizers in

its neighborhood; otherwise, the quantization suffers from code collapse. A simple way to achieve these objectives simultaneously is to impose a loss penalizing the deviation of the KNNs from each vertex (Figure 1C):

$$\mathcal{L}_{\text{reg}} = \mathcal{L}_{\text{KNN}} := (MK)^{-1} \sum_{m=1}^{M} \sum_{k=1}^{K} \mathcal{D}(\mathbf{e}_m, \mathbf{p}^{(m,k)}) \tag{13}$$

where $\mathbf{p}^{(m,k)}$ denotes the $k$-th nearest neighbor of the simplex vertex (onehot vector) $\mathbf{e}_m$ according to a distance/deviation metric $\mathcal{D}$. Two options for $\mathcal{D}$ are considered in this study: the squared L2 distance, $\|\mathbf{e}_m - \mathbf{p}^{(m,k)}\|^2$, and cross-entropy, $-\log p_m^{(m,k)}$.

At first glance, the proposed regularization may appear similar to the commitment and codebook losses used in the gradient-estimation approaches (Equation 10), Both involve nearest neighbors, but the two methods differ in their choice of anchors and neighbors. While commitment and codebook losses take data points as anchors and identify their nearest codebook entries, the proposed method uses codebook entries as anchors and treats data as neighbors. Accordingly, the proposed regularization ensures that every codebook entry receives optimization feedback, whereas commitment/codebook losses may leave some entries untrained if they never become the nearest neighbor of any data point.

A further advantage of the proposed regularization is that Gumbel-softmax sampling is no longer required; smoothed quantizers can be obtained directly as $\mathbf{p} = \boldsymbol{\pi} = \text{softmax}(Q^\mathsf{T}\mathbf{z})$.[2] At the same time, the proposed regularization is fully compatible with Gumbel-softmax sampling; one can simply replace $\mathbf{p}$ (Gumbel-softmax samples) in Equation 13 with $\boldsymbol{\pi}$ (assignment probabilities).

During inference, hard quantization is applied by taking $\text{argmax}_m p_m$ in the onehot representation. In the following section, both the deterministic and stochastic approaches are evaluated on representative benchmarks.

## 4 EXPERIMENTS

The proposed regularization for smoothed vector quantization was benchmarked on two tasks: discrete autoencoding (§4.1) and contrastive learning (§4.2). The Python code used for these experiments is available as supplementary material.

### 4.1 DISCRETE AUTOENCODING

The first benchmark assessed the proposed regularization in the context of discrete autoencoding on the ImageNet dataset (Deng et al., 2009). Input images were convolutionally encoded into latent feature maps, whose pixels were then quantized (Esser et al., 2021; Fifty et al., 2025). The decoder convolutional network reconstructed the input images from these quantized feature maps, and the entire model was trained to minimize the L2 reconstruction loss ($L_{\text{main}}$ in Equation 9). Further details about the network architecture and training setup are provided in Appendix B.1.

Extending prior work (Esser et al., 2021; Fifty et al., 2025), three different combinations of feature map and codebook sizes were examined. In addition to the previously used settings of $H \times W \times C = 16 \times 16 \times 32$ ($M = 1024$) and $64 \times 64 \times 3$ ($M = 8196$), an additional configuration increased the channel dimensionality of the latter to 32 ($H \times W \times C = 64 \times 64 \times 32, M = 8196$).

Each model was trained using four GPUs, and the proposed method computed the $K/4$-nearest neighbors of each simplex vertex per GPU (see §5.3 for further discussion). The number of neighbors was set to $K/4 = 8$ for the L2 distance and $K/4 = 1$ for the cross-entropy otherwise specified.[3] The weight $\beta$ on the commitment loss (Equation 10) in STE (including SimVQ) and rotational gradient estimation was set to 1.0, following Fifty et al. (2025) and Zhu et al. (2025).

Table 1 reports the codebook usage and reconstruction quality scores—including root mean squared error (rMSE), Inception Score (IS; Salimans et al., 2016), and Fréchet Inception Distance (FID;

---

[2]The exact implementation of $\mathbf{p}$ and $\boldsymbol{\pi}$ involves normalization and rescaling; see Appendix A.1 for details.

[3]The value of $K$ was upper-bounded at $8 \times 4$ by available computational resources; With a maximum batch size of 64, the total number of latent pixels was $64 \times 64 \times 64 = 8 \times 4 \times 8196$, allowing only $8 \times 4$ neighbors per vertex of $\Delta^{8196}$. This implementation constraint is further discussed in §5.3.

Table 1: Performance of discrete autoencoding on the ImageNet validation set. Reported metrics are codebook usage and reconstruction quality scores: root mean squared error (rMSE), Fréchet Inception Distance (FID), and Inception Score (IS). The proposed method is denoted as "KNN-L2/CE". Best scores across all methods are highlighted in boldface, while underlined values indicate the best-performing number of nearest neighbors among $K/4 \in \{1, 2, 4, 8\}$.

| | Method | $K/4$ | Feature Map Size; Codebook Size | | | | | | | | | | |
| | | | $16\times16\times32$; 1024 | | | | $64\times64\times3$; 8196 | | | | $64\times64\times32$; 8196 | | | |
| | | | Code Use (↑) | rMSE (↓) | FID (↓) | IS (↑) | Code Use | rMSE | FID | IS | Code Use | rMSE | FID | IS |
| STE | Euclid | — | 4.5% | 0.404 | 124.86 | 36.61 | 100.0% | **0.167** | **7.25** | **402.84** | 1.7% | 0.235 | 22.02 | 290.24 |
| | Cosine | — | 3.0% | 0.381 | 117.95 | 41.48 | 70.9% | 0.197 | 13.97 | 348.31 | 2.8% | 0.186 | 12.11 | 363.86 |
| | SimVQ | — | **100.0%** | 0.340 | 87.26 | 71.33 | **100.0%** | 0.170 | 7.44 | 400.41 | **100.0%** | **0.148** | 3.97 | 436.29 |
| RE | Euclid | — | 3.1% | 0.460 | 170.30 | 19.25 | 78.85% | 0.171 | 10.21 | 377.56 | 0.5% | 0.271 | 40.58 | 203.52 |
| | Cosine | — | 2.8% | 0.423 | 157.77 | 24.33 | 99.5% | 0.194 | 14.67 | 344.17 | 4.4% | 0.180 | 10.83 | 372.89 |
| H-Gmb | PPL | — | **100.0%** | 0.349 | 100.29 | 54.04 | **100.0%** | 0.189 | 19.07 | 321.16 | **100.0%** | **0.163** | 10.33 | 384.04 |
| | **KNN-L2** | 8 | 52.1% | 0.344 | 98.98 | 59.66 | 99.9% | 0.222 | 25.00 | 280.97 | 23.6% | 0.185 | 10.72 | 376.35 |
| | **KNN-CE** | 1 | **100.0%** | 0.368 | 86.49 | 68.27 | **100.0%** | 0.226 | 14.27 | 342.64 | **100.0%** | 0.173 | 3.17 | 435.04 |
| S-Gmb | PPL | — | 55.0% | 0.826 | 173.59 | 9.61 | **100.0%** | 0.183 | 10.28 | 377.79 | 48.4% | 0.386 | 45.61 | 167.14 |
| | **KNN-L2** | 8 | 96.3% | 0.358 | 74.00 | 85.36 | **100.0%** | 0.201 | 10.17 | 373.83 | **100.0%** | 0.193 | 5.73 | 404.18 |
| | **KNN-CE** | 1 | **100.0%** | 0.569 | 194.74 | 16.60 | **100.0%** | 0.233 | 15.41 | 329.78 | **100.0%** | 0.186 | 3.83 | 425.33 |
| Softmax | PPL | — | **100.0%** | 1.112 | 309.60 | 4.55 | 81.6% | 0.701 | 37.99 | 196.72 | 99.8% | 0.725 | 83.07 | 77.98 |
| | **KNN-L2** | 1 | **100.0%** | 0.358 | 76.29 | 81.61 | **100.0%** | 0.224 | 12.08 | 354.36 | **100.0%** | 0.387 | 32.20 | 210.61 |
| | | 2 | **100.0%** | 0.366 | 81.69 | 71.75 | **100.0%** | 0.226 | 12.04 | 348.48 | **100.0%** | 0.175 | 3.62 | 427.42 |
| | | 4 | 100.0% | 0.379 | 87.05 | 62.37 | **100.0%** | 0.205 | 9.01 | 383.56 | **100.0%** | 0.196 | 5.64 | 403.06 |
| | | 8 | 99.9% | **0.343** | 73.72 | **88.53** | 100.0% | 0.199 | 12.64 | 361.23 | **100.0%** | 0.204 | 7.91 | 381.65 |
| | **KNN-CE** | 1 | **100.0%** | 0.366 | 79.80 | 74.69 | **100.0%** | 0.225 | 14.08 | 345.56 | **100.0%** | 0.175 | **2.81** | **437.72** |
| | | 2 | **100.0%** | 0.380 | 106.05 | 50.75 | **100.0%** | 0.227 | 14.34 | 338.37 | **100.0%** | 0.187 | 4.16 | 418.57 |
| | | 4 | 100.0% | 0.748 | 255.59 | 9.03 | **100.0%** | 0.228 | 16.73 | 326.78 | **100.0%** | 0.204 | 6.98 | 387.88 |
| | | 8 | **100.0%** | 1.240 | 467.86 | 3.01 | 100.0% | 0.219 | 14.24 | 343.23 | **100.0%** | 0.199 | 5.27 | 404.10 |

Table 2: Tightness of softmax-based smoothing (without Gumbel sampling), measured by the individual perplexity of individual smoothed quantizers, $\exp(-\sum_{m=1}^{M} p_m \log p_m)$. Reported values are the 75th, 90th, and 99th percentiles, as well as the maximum, computed across all feature-map pixels in the ImageNet validation set.

| Method | $K/4$ | Feature Map Size; Codebook Size | | | | | | | | | | | |
| | | $16\times16\times32$; 1024 | | | | $64\times64\times3$; 8192 | | | | $64\times64\times32$; 8192 | | | |
| | | 75% | 90% | 99% | Max | 75% | 90% | 99% | Max | 75% | 90% | 99% | Max |
| PPL | — | 910.40 | 933.12 | 963.29 | 995.91 | 7559.89 | 7561.99 | 7563.10 | 7565.02 | 7230.60 | 7267.06 | 7315.62 | 7424.16 |
| KNN-L2 | 1 | 1.17 | 1.59 | 2.36 | 6.26 | 1.00 | 1.00 | 1.00 | 4.00 | 2.24 | 40.45 | 86.53 | 249.64 |
| | 2 | 1.00 | 1.08 | 1.86 | 27.37 | 1.00 | 1.00 | 1.06 | 4.65 | 1.14 | 1.55 | 2.33 | 7.92 |
| | 4 | 1.00 | 1.06 | 1.86 | 62.99 | 1.00 | 1.00 | 1.00 | 4.61 | 1.00 | 1.06 | 1.80 | 20.42 |
| | 8 | 1.00 | 1.10 | 1.90 | 120.68 | 3.33 | 3.87 | 5.22 | 12.03 | 1.00 | 1.05 | 1.79 | 107.70 |
| KNN-CE | 1 | 1.18 | 1.60 | 2.38 | 6.28 | 3.96 | 4.51 | 5.81 | 11.96 | 1.09 | 1.45 | 2.17 | 5.94 |
| | 2 | 1.23 | 1.72 | 2.73 | 8.07 | 4.04 | 4.66 | 6.16 | 11.77 | 1.06 | 1.38 | 2.08 | 6.15 |
| | 4 | 1.42 | 1.78 | 2.38 | 5.45 | 4.16 | 4.86 | 6.57 | 13.04 | 1.08 | 1.44 | 2.16 | 6.58 |
| | 8 | 2.38 | 2.80 | 3.71 | 7.78 | 4.46 | 5.31 | 7.39 | 13.81 | 1.29 | 1.75 | 2.65 | 9.26 |

Heusel et al., 2017)–for the proposed method ("KNN-L2/CE") and the baselines. Reconstruction through softmax-smoothed quantization (i.e., without Gumbel randomness) deteriorated substantially when combined with the perplexity-based regularization ("PPL"; Equation 11). This degradation stems from the mismatch between soft quantization during the training and hard quantization at inference; the smoothed quantizers deviated from the simplex vertices, as reflected in their high individual perplexity, $\exp(-\sum_{m=1}^{M} p_m \log p_m)$ (Table 2). These observations support the argument made in the Introduction that perplexity-based regularization alone does not promote tight smoothing.

The perplexity-based regularization was only effective when combined with the Gumbel-softmax sampling, and onehot quantization in the forward computation by STE was necessary to ensure full

codebook usage ("H̲ard-G̲um̲bel"). Otherwise, code collapse was not prevented when the channel dimensionality was large ($C = 32$; "S̲oft-G̲um̲bel").

By contrast, the proposed KNN-based regularization successfully prevented code collapse and achieved near-complete codebook utilization without resorting to Gumbel-softmax sampling. The reconstruction quality was also superior or competitive with the Gumbel-softmax + perplexity approach across all feature map sizes.

The choice of deviation metric (L2 distance vs. cross-entropy, CE) did not result in consistent global superiority, although noticeable code collapse occurred when KNN-L2 regularization was combined with hard Gumbel-softmax sampling, which nonetheless had limited effect on the reconstruction quality.

In terms of computational efficiency, however, cross-entropy proved more favorable. Notably, its performance was highest when the number of neighbors was minimal ($K/4 = 1$) across all feature map sizes (indicated by the underlined scores in Table 1); increasing $K$ did not yield further improvements. Although most smoothed quantizers remained unregularized under this setting, they still achieved low individual perplexities after the training (Table 2), indicating a tight approximation of quantization. By contrast, minimizing only the L2 distance between the simplex vertices and their single nearest neighbor per GPU led to performance degradation for the feature map of size $64 \times 64 \times 8192$. Since requiring more neighbors necessitates a larger batch size (see §5.3 for details), cross-entropy emerges as a more scalable option.

Finally, both STE and rotational gradient estimation (RE) exhibited severe code collapse when the channel dimensionality was large ($C = 32$), reaffirming prior findings that reducing channel dimensionality is necessary for stable training (Yu et al., 2022a; 2024; Mentzer et al., 2024). By contrast, SimVQ maintained full codebook usage across all settings and achieved strong reconstruction quality even at high dimensionality. However, the next section presents a case study in which all gradient-estimation approaches—including SimVQ—encounter code collapse, highlighting the difficulty of achieving robust prevention of code collapse across different settings.

## 4.2 Contrastive Learning

The second experiment evaluated vector quantization methods within Wav2Vec 2.0 pretraining for speech feature extraction (Baevski et al., 2020b). Unlike autoencoding, this pretraining integrates vector quantization directly into the main loss function.

Two codebook configurations were investigated. The first used a single codebook of size 1024. By contrast, the second followed the original work of Baevski et al. (2020b), combining two smaller codebooks—each of size 320—to implement rich code diversity efficiently via *product quantization* (Jégou et al., 2011, see §5.3 for more information). The dimensionality of the codebook vectors was set to 256 for the single-codebook configuration and to 128 for the dual-codebook configuration. The weight $\beta$ on the commitment loss in both STE (including SimVQ) and rotational gradient estimation was set to 1.0 (Fifty et al., 2025; Zhu et al., 2025).

Models were trained on the LibriSpeech dataset, combining all training splits (train-clean-100 + train-clean-360 + train-other-500; Panayotov et al., 2015). Further details on the network architecture and learning objective are provided in Appendix B.2.

Table 3 reports codebook usage. The perplexity-based regularization failed to prevent code collapse in both single- and dual-codebook settings. Likewise, the STE and rotational gradient estimation approaches exhibited the same failure. Remarkably, SimVQ—despite achieving full codebook usage in the discrete autoencoding experiments—offered no observable benefit in this learning paradigm.

By contrast, the proposed KNN-based regularization ensured (near-)complete code utilization in both conditions when cross-entropy was used as the divergence metric on the simplex. As in the discrete autoencoding experiments, a single neighbor per GPU was sufficient to obtain this effect. The L2 metric, on the other hand, proved insufficient to prevent code collapse, especially when combined with Gumbel sampling.

Table 3: Codebook usage in Wav2Vec 2.0 pretraining, evaluated on the LibriSpeech dev-clean split (similar results were observed for the other dev/test splits). The proposed method is denoted as "KNN-L2/CE".

| | | | #Codebooks × Codebook Size | | |
| | | | $1\times1024$ | $2\times320$ | |
| Method | | $K/4$ | | Codebook#1 | Codebook#2 |
| --- | --- | --- | --- | --- | --- |
| STE | Euclid | — | 0.8% | 0.9% | 0.9% |
| | Cosine | ∼ | 0.2% | 0.6% | 0.6% |
| | SimVQ | ∼ | 0.2% | 0.6% | 0.6% |
| RE | Euclid | — | 2.5% | 0.6% | 0.6% |
| | Cosine | ∼ | 0.2% | 0.6% | 0.6% |
| H-Gmb | PPL | ∼ | 0.7% | 0.6% | 0.6% |
| | **KNN-L2** | 2 | 0.2% | 0.6% | 0.6% |
| | **KNN-CE** | 2 | 99.7% | **100.0%** | **100.0%** |
| S-Gmb | PPL | — | 0.3% | 0.6% | 0.6% |
| | **KNN-L2** | 2 | 90.1% | 0.6% | 0.6% |
| | **KNN-CE** | 2 | **100.0%** | **100.0%** | **100.0%** |
| Softmax | PPL | ∼ | 0.2% | 1.2% | 1.2% |
| | **KNN-L2** | 1 | 82.4% | **100.0%** | **100.0%** |
| | | 2 | 60.4% | **100.0%** | **100.0%** |
| | **KNN-CE** | 1 | 99.5% | **100.0%** | **100.0%** |
| | | 2 | **100.0%** | **100.0%** | **100.0%** |

## 5 DISCUSSIONS

### 5.1 SUMMARY OF FINDINGS & CONTRIBUTIONS

This study introduced a simple and unified regularization method that simultaneously tightens smoothed vector quantization and promote effective code utilization. The proposed method successfully prevented code collapse in two representative applications of vector quantization: a middle layer in discrete autoencoding (§4.1) and target construction in contrastive learning (§4.2). This robustness is noteworthy, as prior approaches were effective only in specific settings and remained vulnerable to code collapse in others.

The proposed method is geometrically intuitive and straightforward, yet appears unaddressed in the existing literature. Research on neural vector quantization has traditionally been rooted in variational autoencoding (Kingma & Welling, 2014), primarily aiming to extend this stochastic framework to discrete variables (Jang et al., 2017; van den Oord et al., 2017). The issue of code collapse was recognized (or documented) later, and workarounds were developed independently of the quantization methods themselves (Dieleman et al., 2018; Baevski et al., 2020b; Dhariwal et al., 2020).

By contrast, the present work reformulates neural vector quantization as a simple smoothing problem: onehot vectors are approximated by elements of the simplex. Within this perspective, concentrating the approximators near the simplex vertices naturally arises as a desirable property. This reformulation, together with the proposed regularization strategy, represents a key conceptual contribution of the study.

### 5.2 ALTERNATIVE IMPLEMENTATIONS OF THE INTENDED REGULARIZATION

Alternative regularization strategies could also achieve the intended distribution of smoothed quantizers ($\mathbf{p}$ or $\boldsymbol{\pi}$) around the simplex vertices. For example, one could align smoothed quantizers with a Dirichlet distribution with concentration parameters $\alpha_1 = \cdots = \alpha_K < 1.0$ (Figure 2). The probability density of such a distribution is highest at the vertices of the simplex, thus matching the ideal distribution of smoothed quantizers. A possible formalization of this alignment is based on the Kullback-Leibler (KL) divergence between the Dirichlet prior ($\mathbb{P}$) and the distribution inferred from

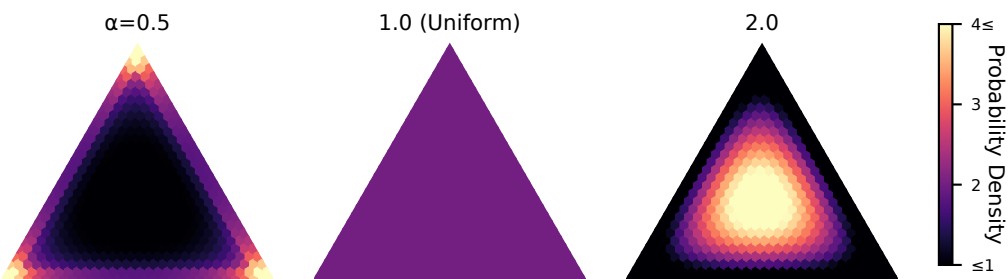

Figure 2: Dirichlet distributions on the simplex $\Delta^{3-1}$ with concentration parameters $\alpha_1 = \alpha_2 = \alpha_3 = \alpha$, where $\alpha \in \{0.5, 1.0, 2.0\}$.

smoothed quantizers ($\mathbb{Q}$).

$$\mathcal{D}_{\text{KL}}\left(\mathbb{P} \mid \mathbb{Q}\right) := \int_{\Delta^{M-1}} \mathbb{P}(\mathbf{p}) \log \frac{\mathbb{P}(\mathbf{p})}{\mathbb{Q}(\mathbf{p})} d\mathbf{p} \tag{14}$$

However, Equation 14 is difficult to use directly as a regularization loss, due to the complexity of estimating $\mathbb{Q}$ from sample quantizers, $\mathbf{p}$. Although one could constrain $\mathbb{Q}$ as another Dirichlet to make the KL divergence tractable, maximum likelihood estimation of its parameters requires iterative algorithms (e.g., the Newton-Raphson method; Ronning, 1989; Sklar, 2014; Wicker et al., 2008), complicating and slowing gradient-based optimization in deep learning frameworks. Moreover, this estimation involves digamma and trigamma functions (Sklar, 2014), whose derivatives can explode when the concentration parameters approach small values—as desired for tight smoothing—during the course of learning.

A more practical approach is to approximate Equation 14 itself in a tractable manner. For instance, Perez-Cruz (2008) proposed a KNN-based estimation of the KL divergence that relies solely on *samples* from the two distributions, using the $k$-th nearest neighbor from $\mathbb{Q}$ for each sample from $\mathbb{P}$. The regularization method proposed in this study can thus be interpreted as minimizing this estimated KL divergence, with the samples from $\mathbb{P}$ constrained to onehot vectors.

### 5.3 LIMITATIONS

A primary limitation of the proposed method is its memory requirement. When training across multiple GPUs, the KNN-based regularization identifies $K$ nearest smoothed quantizers ($\mathbf{p}$ or $\boldsymbol{\pi}$) per simplex vertex *on each GPU*, rather than finding global neighbors across all GPUs. Consequently, each GPU must have sufficient VRAM to store at least $KM$ latent pixels/frames, where $M$ denotes the codebook size. This requirement can become prohibitive when $M$ is large (i.e., for fine-grained quantization), although empirically, a single neighbor per GPU appeared sufficient to prevent code collapse when cross-entropy was used as the divergence metric.

One possible workaround is to randomly select a subset of simplex vertices when computing the regularization loss, rather than using all vertices in a single iteration. In expectation, this achieves the same effect as the original implementation, although its empirical effectiveness remains to be assessed in future studies.

Additionally, fine-grained quantization can be achieved more efficiently using smaller $G$ codebooks in combination (product quantization; Jégou et al., 2011), as explored in the Wav2Vec 2.0 pre-training (§4.2). This approach represents $\prod_{g=1}^{G} M_g$ distinct quantized vectors while requiring only $K \sum_{g=1}^{G} M_g$ smoothed quantizers per GPU. Leveraging these strategies, the proposed method can overcome its limitation and become applicable to real-world scenarios.

### ACKNOWLEDGMENTS

Hidden for double-blind review.

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

## A  IMPLEMENTATION OF THE QUANTIZATION METHODS

This section provides details on the implementation of the quantization methods.

Table 4: Hyperparameters for discrete autoencoding.

| | $16 \times 16 \times 32$ | $64 \times 64 \times 3$ | $64 \times 64 \times 32$ |
|---|---|---|---|
| Feature Map Size | $16 \times 16 \times 32$ | $64 \times 64 \times 3$ | $64 \times 64 \times 32$ |
| Codebook Size | 1024 | 8196 | 8196 |
| Latent Channels | $128 \to 128 \to 64 \to 64 \to 32 \to 32$ | $128 \to 64 \to 32 \to 3$ | $128 \to 64 \to 32 \to 32$ |
| Height & Width | $256 \to 128 \to 64 \to 32 \to 16 \to 16$ | $256 \to 128 \to 64 \to 64$ | $256 \to 128 \to 64 \to 64$ |
| Batch Size | 64 | 64 | 64 |
| Training Epochs | 25 | 20 | 20 |
| Warmup Iterations | 16,000 | 16,000 | 16,000 |

### A.1 SMOOTHED QUANTIZATION

Smoothed quantizers $\mathbf{p}$ were computed as $\mathbf{p} = \mathrm{softmax}(\hat{Q}^\mathsf{T}\hat{\mathbf{z}}/t)$. In other words, the codebook vectors $(\mathbf{q}_1, \ldots, \mathbf{q}_M) = Q$ and the feature vectors $\mathbf{z}$ were first L2-normalized, and their product (i.e., cosine similarity) was rescaled by a learnable temperature $t$. This temperature was shared across the codebook so that all the logits had the same amplitude. Assignment probabilities $\pi$ for Gumbel-softmax sampling were computed in the same way, while the additional temperature parameter—$\tau$ in Equation 7—was fixed as 1.0.[4]

### A.2 HARD QUANTIZATION

The weight $\beta$ on the commitment loss in Equation 10 was set to 1.0 (Fifty et al., 2025; Zhu et al., 2025), based on the previous observations that its value does not significantly affect learning outcomes within the range 0.1–2.0. (van den Oord et al., 2017).

## B DETAILS OF THE EXPERIMENTS

### B.1 DISCRETE AUTOENCODING

This section provides implementation details for the autoencoding experiment described in §4.1.

The network architecture followed prior work on discrete autoencoding of ImageNet (Esser et al., 2021; Fifty et al., 2025). Input images were center-cropped to $H \times W \times C = 256 \times 256 \times 3$. The encoder first expanded the channel dimensionality of the input images from 3 to 256 by convolution, and then progressively downsampled them through a series of strided convolutions (see Table 4 for the spatial and channel sizes at each layer). Each downsampling layer was followed by a residual block (He et al., 2016). The decoder reconstructed the input images by upsampling the latent feature maps with a sequence of interpolations and residual blocks.

To improve memory efficiency—particularly important for the proposed KNN-based regularization—all but the input and output layers were implemented as depthwise separable convolutions (Chollet, 2017). All convolutional kernels had size $3 \times 3$.

Training employed the AdamW optimizer with $(\beta_1, \beta_2) = (0.9, 0.99)$ and a weight decay coefficient of $10^{-4}$, except for Euclidean-based STE/rotational hard quantization, where weight decay was set to zero. The learning rate was linearly warmed up from 0.0 to $\rho_{\max}$, and subsequently annealed to $0.5\rho_{\max}$ by cosine scheduling. The maximum learning rate $\rho_{\max}$ was set to $5 \times 10^{-5}$ for Euclidean-based STE/rotational hard quantization, and to $10^{-4}$ for all other configurations (Fifty et al., 2025).

Inception Score (IS; Salimans et al., 2016) and Fréchet Inception Distance (FID; Heusel et al., 2017) were estimated using the ImageNet-pretrained Inception V3 provided in torchvision.

### B.2 WAV2VEC 2.0

This section provides details for the Wav2Vec 2.0 pretraining discussed in §4.2.

---

[4]Previous studies manually annealed the Gumbel-softmax temperature from $\tau = 0.5$ to 2.0, scaling it by 0.999995 at each iteration (Baevski et al., 2020b). This approach was also tested in the experiments here but did not yield improvements over the fixed temperature.

Table 5: Hyperparameters for Wav2Vec 2.0.

| | | |
|---|---|---|
| Input Frequency | 16kHz | |
| Latent Frequency | 50Hz | |
| Codebook Dimensionality | 256 | |
| Codebook Size | 1024 | |
| **CNN** | | |
| Latent Channels | 512 | |
| Kernel Sizes | $10 \to 3 \to 3 \to 3 \to 3 \to 2 \to 2$ | |
| Strides | $5 \to 2 \to 2 \to 2 \to 2 \to 2 \to 2$ | |
| **Transformer** | | |
| # Layers | 12 | |
| Model Dimensionality | 768 | |
| # Heads | 8 | |
| Feed-Forward Dimensionality | 4096 | |
| Dropout Rate | 0.1 | |
| Layer Drop | 0.05 | |
| | #Codebooks×Codebook Size | |
| | $1 \times 1024$ | $2 \times 320$ |
| Batch Size | 128 | 64 |
| Training Epochs | 128 | 20 |
| Warmup Iterations | 32,000 | 10,000 |

The model consisted of a convolutional feature encoder followed by a Transformer module (Baevski et al., 2020b). The convolutional encoder extracted latent feature sequences from input waveforms (16kHz→50Hz). Then, a subset of these latent vectors was masked and fed into the Transformer, whose outputs $\mathbf{y}_t$ were trained to predict the quantized version $\mathbf{q}_t$ of the masked vectors. The masking scheme followed Baevski et al. (2020b); random 6.5% of the latent vectors were masked, together with the following 10 time steps. The learning objective was:

$$\mathcal{L}_{\text{main}} = -\log \frac{\exp(\hat{\mathbf{y}}_t \hat{\mathbf{q}}_t^{\mathsf{T}} / \mathcal{T})}{\sum_{\hat{\mathbf{q}} \sim \mathcal{Q}} \exp(\hat{\mathbf{y}}_t \hat{\mathbf{q}}^{\mathsf{T}} / \mathcal{T})} \tag{15}$$

where $\hat{\cdot}$ denotes the L2-normalization of vectors (i.e., measuring the cosine similarity), and $\mathcal{T} := 0.1$ is the temperature parameter. For each masked vector, a set of distractors $\tilde{\mathbf{q}}$ was sampled from the other quantized vectors according to a distribution $\mathcal{Q}$.

Both the convolutional encoder and Transformer were implemented using the publicly available code in torchaudio, and only the quantization module was implemented from scratch. All components are provided in the supplementary material.

Input waveforms were randomly cropped to the length of 250k samples. Both stages employed the AdamW optimizer with $(\beta_1, \beta_2) = (0.9, 0.99)$ and zero weight decay. The learning rate was warmed up from 0.0 to $5.0 \times 10^{-4}$, and then annealed to $5.0 \times 10^{-6}$ by cosine scheduling.

When the single-codebook condition was first examined, training was run for 128 epochs, following the original schedule (Baevski et al., 2020b). However, since convergence occurred rapidly, the number of epochs was reduced in the dual-codebook condition to improve time efficiency.

