# OpenReview forum: "Pushing Toward the Simplex Vertices: A Simple Remedy for Code Collapse in Smoothed Vector Quantization"
_ICLR.cc/2026/Conference — Submitted to ICLR 2026_

### Official Review · Reviewer_249k · 2025-10-21

**Soundness:** 3
**Presentation:** 3
**Contribution:** 3
**Rating:** 6
**Confidence:** 3

**Summary:**

This paper addresses the non-differentiability of hard quantization by smoothing assignments while also aiming to avoid code collapse. It reformulates VQ as smoothing of one-hot vectors and adds a simple KNN regularization that pulls smoothed assignments toward all simplex vertices (using L2 or cross-entropy), compatible with softmax or Gumbel-softmax. On ImageNet autoencoding and wav2vec2 pretraining, the method shows reasonable performance against the reported baselines.

**Strengths:**

- The paper is well written. The technical parts are clear and detailed.
- Smoothing the VQ process is an interesting idea. It provides new insight and motivates a simple KNN regularization to reduce code collapse. The loss is simple and effective.
- Experiments confirm the effectiveness of the proposed method across tasks.

**Weaknesses:**

- First, the experimental scale is limited, and the paper mostly compares its own variants. There are few comparisons to recent methods, and the related work leans on pre‑2023 citations, with the rotation trick as the only recent item. As a result, the basis feels thin for an active area like VQ codebook learning.
- Second, more experiments are needed to study how batch size influences the KNN regularization. For example, compute the loss using groups within the same batch and analyze the trends. This would show whether the method scales to larger settings.
- Third, include an ablation of the Gumbel‑Softmax temperature to make the study more complete and to clarify its interaction with the proposed loss.

**Questions:**

- Please refer to the Weaknesses section. Overall I appreciate the technical contribution, but because the experiments appear insufficient and the discussion of recent related work is limited, I will keep the current score.

---

> ### Author Response · Authors · 2025-11-17
> **Progress in additional experiments**
>
> Thank you for the constructive and thoughtful feedback.
>
> Based on the overall comments we received, we have first examined the effect of varying $K$ on model performance, and the corresponding results are summarized in Table 1. We are now preparing an additional baseline comparison with SimVQ (Zhu et al., 2025, ICCV) to directly address the concern regarding the limited set of recent methods. We will submit an updated rebuttal once these results are obtained. Thank you for your patience.
>
> Regarding your third question (on the temperature parameter), we tested both a fixed temperature $\tau=1.0$ and the annealing schedule proposed by Baevski et al. (2020, NeurIPS). No noticeable differences were observed between the two approaches. Please refer to Appendix A.1, in particular Footnote 4, for details.

---

> > ### Author Response · Authors · 2025-11-26
> > **Explored SimVQ**
> >
> > We have completed the evaluation of SimVQ. In the discrete autoencoding experiments, SimVQ achieved full codebook usage and delivered strong reconstruction quality, although it did not outperform the best methods in any setting. In the wav2vec contrastive learning experiments, however, SimVQ provided no observable improvement; it suffered from code collapse in the same manner as the canonical STE and rotational gradient estimation. These results underscore the value of our proposed approach, which offers a new remedy for code collapse working that remains effective precisely in settings where existing methods (including SimVQ) fail.

---

> > > ### Author Response · Authors · 2025-12-03
> > > **A remark regarding the batch-size ablation (in response to weakness/question #2)**
> > >
> > > In the revised manuscript, we have assessed the effect of varying $K$ on model performance, and summarized the results in Table 1. We observed the following:
> > > - **Even the minimum setting of $K/4=1$ (i.e., one neighbor per GPU) was sufficient to prevent code collapse**. Although most latent data points remained unregularized under this configuration, they eventually clustered around the simplex vertices by the end of training.
> > > - Moreover, **CE performed best in this minimal setting in terms of reconstruction quality in discrete autoencoding (i.e., increasing $K$ led to degraded performance).** This is encouraging in practice, as the smallest $K$ also minimizes the computational cost of the proposed method.
> > > - By contrast, the L2 distance did not exhibit a consistently superior choice of $K$, and no interpretable relationship emerged between best-performing $K$ and the feature-map resolution.
> > >
> > > We believe these additional experiments partially address your question regarding batch-size dependence, since the scalability of our proposed method is governed by the relative sizes of the batch (or the number of latent data points), the codebook ($M$), and $K$. The sufficiency of the minimal setting ($K/$#GPUs$=1$) suggests that the batch size can be reduced as long as the number of latent points remains larger than the codebook size $M$.
> > >
> > > In addition, we would like to note that when $K/$#GPUs$=1$, the proposed regularization has the same order of computational complexity as the standard commitment and codebook losses ($O(NM)$, where $N$ denotes the number of latent data points per GPU and $M$ is the codebook size). The difference lies only in the direction of nearest-neighbor search (across data points vs. across codebook entries). Moreover, computing both the commitment loss and codebook loss (necessary because they have different stop-gradient targets) doubles the cost, whereas the proposed regularization requires only a single pass.
> > >
> > > Therefore, at least from a memory perspective, the proposed regularization would scale similarly to existing methods, although its empirical effectiveness across different batch and/or codebook sizes remains an open question.

---

### Official Review · Reviewer_PY24 · 2025-11-01

**Soundness:** 3
**Presentation:** 3
**Contribution:** 3
**Rating:** 6
**Confidence:** 3

**Summary:**

The paper addresses the issue of code collapse in smoothed vector quantization by proposing a simple K-nearest neighbor (KNN) regularization that simultaneously enforces tight smoothing and uniform code utilization. Empirical evaluations on discrete image autoencoding and contrastive speech representation learning demonstrate that the proposed approach achieves nearly complete codebook utilization and competitive or superior reconstruction quality compared to baselines including straight-through estimation, Gumbel-softmax, and perplexity-based regularization.

**Strengths:**

1. The illustration and explanation of smoothed vector quantization from a simplex perspective provide a clear geometric intuition that unifies and clarifies connections with prior approaches.
2. The proposed KNN regularization is an intuitive idea that offers strong empirical benefits compared to prior entropy or perplexity-based regularization methods.

**Weaknesses:**

1. While the method is designed to improve both smoothing tightness and codebook utilization, the paper does not provide quantitative analysis or visualization to support the claim about improved smoothing tightness.
2. The KNN loss is acknowledged to be memory-inefficient, but the paper does not include any experiments or analysis to quantify the computational overhead or explore mitigation strategies.

**Questions:**

1. How does the proposed regularization scale to very large codebooks (e.g., M = 16k ~ 64k)? Have you explored memory-efficient approximations such as partial vertex sampling or approximate KNN search?
2. How sensitive are the results to the choice of K (number of nearest neighbors)? Is there a principled way to select K relative to the codebook size or batch size?
3. While the paper demonstrates improved code utilization and reconstruction performance, does this lead to improvements in downstream tasks (e.g., image generation or text-to-speech synthesis)?

---

> ### Author Response · Authors · 2025-11-17
> **Investigated effects of varying K**
>
> Thank you for the constructive and thoughtful feedback.
>
> We have first assessed the effect of varying $K$ on model performance, and the results are summarized in Table 1. We found:
> - **Even the minimum setting of $K/4=1$ (i.e., one neighbor per GPU) was sufficient to prevent code collapse**. Although most data points remained unregularized under this configuration, they eventually clustered around the simplex vertices by the end of training.
> - Moreover, **CE performed best in this minimal setting in terms of reconstruction quality in discrete autoencoding (i.e., increasing $K$ led to degraded performance).** This is encouraging in practice, as the smallest $K$ also minimizes the computational cost of the proposed method.
> - By contrast, the L2 distance did not exhibit a consistently superior choice of $K$, and no interpretable relationship emerged between best-performing $K$ and the feature-map resolution.
>
> We have also added an evaluation of smoothing tightness, quantified by the perplexity of individual smoothed quantizers (Table 2).
> The proposed KNN-based regularization indeed yielded low individual perplexities, indicating proximity to simplex vertices and a tight approximation of hard quantization.
> By contrast, the average-PPL maximization baseline increased the perplexity of individual smoothed quantizers, pushing the smoothed assignments toward the interior of the simplex and causing a mismatch with hard quantization at inference.
> This analysis confirms the intuition illustrated in the Introduction (Figure 1).

---

> > ### Author Response · Authors · 2025-12-03
> > **A note on scalability (in response to question #1)**
> >
> > We would like to note that when $K/$#GPUs$=1$ (i.e., when KNN-CE performed best), the proposed regularization has the same order of computational complexity as the standard commitment and codebook losses ($O(NM)$, where $N$ denotes the number of latent data points per GPU and $M$ is the codebook size). The difference lies only in the direction of nearest-neighbor search (across data points vs. across codebook entries). Moreover, computing both the commitment loss and codebook loss (necessary because they have different stop-gradient targets) doubles the cost, whereas the proposed regularization requires only a single pass.
> >
> > Therefore, at least from the perspective of memory, the proposed regularization would scale similarly to existing methods, although its empirical effectiveness for preventing code collapse in very large codebooks remains an open question.

---

### Official Review · Reviewer_Zwav · 2025-11-01

**Soundness:** 3
**Presentation:** 2
**Contribution:** 3
**Rating:** 4
**Confidence:** 4

**Summary:**

This paper proposes a new regularization method for smoothed vector quantization that simultaneously promotes near one-hot assignments and prevents codebook collapse. Their method primarily varies from prior methods in its choice of anchors and neighbours. More specifically, they use codebook entries as anchors and data as neighbours. Using this, the KNN-based regularizer encourages samples to cluster near all vertices, ensuring full codebook utilization and tight smoothing. They test their method on two VQ tasks (auto-encoding and contrastive learning) and show that their method performs on par with or better than the baseline methods in different codebook settings.

**Strengths:**

1. Their method naturally incorporates all the entries of the codebook, which ensures maximum codebook utilization by design.
2. They achieve 100% codebook utilization among all the tasks they train, while performing on par or better than baseline methods
3. The regularization loss they propose is quite flexible and can be combined with other backbones, such as Gumbel softmax, L2 or cross entropy losses, etc.

**Weaknesses:**

1. Missing comparison with other baseline methods such as Group VQ, SimVQ etc.,
2. Since they use the codebook entries as anchors, some of the datapoints might not contribute to the regularization loss
3. Their method does not perform the best in some settings like in table 1, 64x64x3 setting, where the STE method performs the best

**Questions:**

1. The paper might improve by adding more baseline methods to show how their method compares with current state-of-the-art methods
2. How does the method perform on Image generation (diffusion, flow-based, or autoregressive) or language models?
3. How sensitive is the performance of the method in terms of K
4. A dedicated ablation study and an explanation for the choice of hyperparameters can help in reproducibility.

---

> ### Author Response · Authors · 2025-11-17
> **Investigated effects of varying K**
>
> Thank you for the constructive and thoughtful feedback.
>
> We have first assessed the effect of varying $K$ on model performance, and the results are summarized in Table 1. We found:
> - **Even the minimum setting of $K/4=1$ (i.e., one neighbor per GPU) was sufficient to prevent code collapse**. Although most data points remained unregularized under this configuration, they eventually clustered around the simplex vertices by the end of training.
> - Moreover, **CE performed best in this minimal setting in terms of reconstruction quality in discrete autoencoding (i.e., increasing $K$ led to degraded performance).** This is encouraging in practice, as the smallest $K$ also minimizes the computational cost of the proposed method.
> - By contrast, the L2 distance did not exhibit a consistently superior choice of $K$, and no interpretable relationship emerged between best-performing $K$ and the feature-map resolution.
>
> Additionally, we are preparing an evaluation of an additional baseline (SimVQ; Zhu et al., 2025, ICCV). We will submit a further rebuttal once the results are obtained.

---

> > ### Author Response · Authors · 2025-11-26
> > **Explored SimVQ**
> >
> > We have completed the evaluation of SimVQ. In the discrete autoencoding experiments, SimVQ achieved full codebook usage and delivered strong reconstruction quality, although it did not outperform the best methods in any setting. In the wav2vec contrastive learning experiments, however, SimVQ provided no observable improvement; it suffered from code collapse in the same manner as the canonical STE and rotational gradient estimation. These results underscore the value of our proposed approach, which offers a new remedy for code collapse working that remains effective precisely in settings where existing methods (including SimVQ) fail.

---

### Official Review · Reviewer_Ssva · 2025-11-01

**Soundness:** 2
**Presentation:** 3
**Contribution:** 2
**Rating:** 4
**Confidence:** 3

**Summary:**

- This paper proposes a simple yet effective KNN-based regularization method to stabilize smoothed vector quantization. A well-behaved quantizer should produce one-hot–like assignments while maintaining balanced codebook usage. The proposed approach achieves both by penalizing the distance between each simplex vertex and its K nearest soft assignments. This encourages all codebook entries to be actively used while keeping each assignment sharply concentrated near a single vertex.


- As a result, the method mitigates the train-test mismatch that occurs in perplexity-based regularization, where training uses soft assignments but inference requires hard quantization.


- Experiments on discrete image autoencoding and contrastive speech representation learning demonstrate that the proposed regularizer effectively prevents code collapse, achieves near-complete code utilization, and delivers improved reconstruction and representation quality compared with other baselines.

**Strengths:**

- Empirical results support that KNN is superior to Perplexity regularizer, as it effectively reduce the mismatch between soft / hard assignment when training and testing.

- Experiment results in representation learning highlights the robustness of the proposed method when adopted with cross-entropy (KNN-CE), which achieves 100% usage in both single and dual codebooks, where other methods fail.

- The paper is generally clear. The Related Work section provides not only a concise overview of recent advances in vector quantization but also a coherent narrative that situates the proposed method within the broader literature. It effectively summarizes the fundamental components of VQ, such as the quantization step, gradient estimation, and smoothing strategies, making it accessible and informative even to readers new to the topic.

**Weaknesses:**

- The paper lacks an ablation study on the number of neighbors (K), which is a key hyperparameter of the proposed regularizer.


- There is limited qualitative results in the experiment section. Beyond the simplex visualizations, it would be valuable to include additional qualitative examples such as reconstructed images or assignment-map visualizations to illustrate the impact of KNN-based over Perplexity regularization.


- Lack of experimental settings: details such as the exact loss components (reconstruction, codebook, commitment) and weighting used for baselines like STE, RE, and Perplexity regularization should be declared.

**Questions:**

- The experimental results show that the proposed method combined with L2 distance performs noticeably worse than KNN + CE, particularly on the contrastive learning task. My main concern is about the method’s robustness to the choice of distance metric:

Is the proposed regularizer sensitive to the form of the distance function?

Could you provide theoretical intuition or gradient-level analysis explaining why CE behaves better than L2 in contrastive settings?

Additional experiments or ablation studies comparing both distances under controlled settings would strengthen the claim.

- Please provide further analysis on the influence of number of neighbors (K):

When K is small, can the method still mitigate the soft/hard assignment mismatch observed in Perplexity regularization?

Please include an ablation study on varying K and compare it with other baselines to measure effectiveness of the proposed method.

- Because the value of K directly affects GPU memory usage, please report experiments for both baselines and the proposed method under similar GPU resource.

---

> ### Author Response · Authors · 2025-11-17
> **Investigated effects of varying K**
>
> Thank you for the constructive and thoughtful feedback.
>
> We have first assessed the effect of varying $K$ on model performance, and the results are summarized in Table 1. We found:
> - **Even the minimum setting of $K/4=1$ (i.e., one neighbor per GPU) was sufficient to prevent code collapse**. Although most data points remained unregularized under this configuration, they eventually clustered around the simplex vertices by the end of training.
> - Moreover, **CE performed best in this minimal setting in terms of reconstruction quality in discrete autoencoding (i.e., increasing $K$ led to degraded performance).** This is encouraging in practice, as the smallest $K$ also minimizes the computational cost of the proposed method.
> - By contrast, the L2 distance did not exhibit a consistently superior choice of $K$, and no interpretable relationship emerged between best-performing $K$ and the feature-map resolution.
>
> These additional results also address the question regarding CE vs. L2; The proposed regularizer is indeed sensitive to the choice of divergence metric, and CE appears more robust across tasks and more computational efficient.
>
> We have also added an evaluation of smoothing tightness, quantified by the perplexity of individual smoothed quantizers (Table 2).
> The proposed KNN-based regularization indeed yielded low individual perplexities, indicating proximity to simplex vertices and a tight approximation of hard quantization.
> By contrast, the average-PPL maximization baseline increased the perplexity of individual smoothed quantizers, pushing the smoothed assignments toward the interior of the simplex and causing a mismatch with hard quantization at inference.
> This analysis confirms the intuition illustrated in the Introduction (Figure 1).
>
> Additionally, we are preparing an evaluation of an additional baseline (SimVQ; Zhu et al., 2025, ICCV). We will submit a further rebuttal once the results are obtained.

---

> > ### Author Response · Authors · 2025-11-26
> > **Evaluated SimVQ**
> >
> > We have completed the evaluation of SimVQ. In the discrete autoencoding experiments, SimVQ achieved full codebook usage and delivered strong reconstruction quality, although it did not outperform the best methods in any setting. In the wav2vec contrastive learning experiments, however, SimVQ provided no observable improvement; it suffered from code collapse in the same manner as the canonical STE and rotational gradient estimation. These results underscore the value of our proposed approach, which offers a new remedy for code collapse working that remains effective precisely in settings where existing methods (including SimVQ) fail.
> >
> > We have also clarified that the commitment-loss weight was set to $\beta=1.0$, following Fifty et al. (2025) and Zhu et al. (2025). Although this information was originally provided in Appendix A.2, the revised draft now includes it in the main text (Sections 4.1 and 4.2) for improved visibility.

---

> > > ### Author Response · Authors · 2025-12-03
> > > **A note on the computational cost**
> > >
> > > We would like to note that the computational cost of the proposed KNN regularization is small relative to that of the backbone network. In all experiments, the batch size was set to the maximum feasible value (restricted to powers of two) for the available GPU memory, and this value was invariant across all quantization methods. Thus, the reported results were indeed obtained "under similar GPU resource for both baselines and the proposed method".
> > >
> > > In addition, when $K/$#GPUs$=1$ (i.e., when KNN-CE performed best), the proposed regularization has the same order of computational complexity as the standard commitment and codebook losses ($O(NM)$, where $N$ denotes the number of latent data points per GPU and $M$ is the codebook size). The difference lies only in the direction of nearest-neighbor search (across data points vs. across codebook entries). Moreover, computing both the commitment loss and codebook loss (necessary because they have different stop-gradient targets) doubles the cost, whereas the proposed regularization requires only a single pass.

---

### Author Response · Authors · 2025-12-03
**Summary of Rebuttals**

This post provides a summary of the revisions made in response to the reviewers' comments, to assist AC in their decision.
We would like to thank all the reviewers again for their constructive and thoughtful feedback.
They were all highly valuable in improving the manuscript.

- **Assessment of varying K** (requested by Reviewers Ssva, Zwav, PY24, and also partially addressing a request from Reviewer 249k):
  - Additional experiments were conducted by varying the number K of the nearest neighbors used in the proposed regularization loss.
  - It was found that **even a single neighbor per vertex per GPU (K/4=1) is sufficient to prevent code collapse**.
  - This result **alleviates concerns regarding the required computational resources**.
- **Evaluation of smoothing tightness** (requested by Reviewers Ssva and PY24):
  - In addition to code usage (i.e., assessing code-collapse prevention), the tightness of the smoothed quantization was evaluated using the perplexity of individual quantizers.
  - **Even the minimal setting of K/4=1 resulted in low individual perplexity (i.e., a tight approximation)**.
  - This finding supports the claim that the proposed regularization **simultaneously prevents code collapse and encourages tight approximation**.
- **Evaluation of a more recent baseline** (requested by Reviewers Ssva, Zwav, 249k):
  - An additional baseline model, SimVQ (Zhu, 2025, ICCV), was evaluated.
  - SimVQ achieved strong performance in discrete autoencoding but **exhibited code collapse in wav2vec contrastive learning**.
  - This outcome highlights the robustness of the proposed method, which successfully prevented code collapse across all experimental settings.

---

### Meta-Review · Area_Chair_fq9Y · 2026-01-05

**Summary:**

AC sees that the authors have addressed concerns such as the effect of K, batch size etc, in and of itself related to the proposed method.

However, AC's biggest concern is the lack of validation, not so much the method itself, but the lack of showing this has indeed impact on downstream tasks is the largest weakness of this paper.

Upon weighing the pros and cons, AC urged the authors to run the method on downstream tasks and send the paper to the next venue.

**Reviewer Concerns:**

Addressed most ablations on K, batch size etc, so I think that could have keep or raised some ratings. Did not even address the downstream tasks issue, which is a big concern.

**Reviewer Scores:**

Hard to tell but AC thinks the impact on downstream tasks is completely missing could lower some ratings.

---

### Decision · Program_Chairs · 2026-01-26

Reject